# Phylogeny and Taxonomic Revision of the Genus *Melanosciadium* (Apiaceae), Based on Plastid Genomes and Morphological Evidence

**DOI:** 10.3390/plants13060907

**Published:** 2024-03-21

**Authors:** Qiu-Ping Jiang, Xian-Lin Guo, An-Qi Zhao, Xing Fan, Qing Li, Song-Dong Zhou, Xing-Jin He

**Affiliations:** 1Key Laboratory of Bio-Resources and Eco-Environment of Ministry of Education, College of Life Sciences, Sichuan University, Chengdu 610065, China; jiangqiup@163.com (Q.-P.J.); zaq.gz.01@foxmail.com (A.-Q.Z.); zsd@scu.edu.cn (S.-D.Z.); 2Chengdu Institute of Biology, Chinese Academy of Sciences, Chengdu 610093, China; guoxl1@cib.ac.cn; 3Chengdu Branch of Giant Panda National Park, Chengdu 610083, China; 17780464420@163.com (X.F.); lqgoing@163.com (Q.L.)

**Keywords:** Apiacea, *Angelica tsinlingensis*, *Ligusticum angelicifolium*, *Melanosciadium*, *Melanosciadium jinzhaiensis*, new species

## Abstract

*Melanosciadium* is considered a monotypic genus and is also endemic to the southwest of China. No detailed phylogenetic studies or plastid genomes have been identified in *Melanosciadium*. In this study, the plastid genome sequence and nrDNA sequence were used for the phylogenetic analysis of *Melanosciadium* and its related groups. *Angelica tsinlingensis* was previously considered a synonym of *Hansenia forbesii*. Similarly, *Ligusticum angelicifolium* was previously thought to be the genus *Angelica* or *Ligusticopsis*. Through field observations and morphological evidence, we believe that the two species are more similar to *M. pimpinelloideum* in leaves, umbel rays, and fruits. Meanwhile, we found a new species from Anhui Province (eastern China) that is similar to *M. pimpinelloideum* and have named it *M. Jinzhaiensis*. We sequenced and assembled the complete plastid genomes of these species and another three *Angelica* species. The genome comparison results show that *M. pimpinelloideum*, *A. tsinlingensis*, *Ligusticum angelicifolium*, and *M. jinzhaiensis* have similarities to each other in the plastid genome size, gene number, and length of the LSC and IR regions; the plastid genomes of these species are distinct from those of the *Angelica* species. In addition, we reconstruct the phylogenetic relationships using both plastid genome sequences and nrDNA sequences. The phylogenetic analysis revealed that *A. tsinlingensis*, *M. pimpinelloideum*, *L. angelicifolium*, and *M. jinzhaiensis* are closely related to each other and form a monophyletic group with strong support within the Selineae clade. Consequently, *A. tsinlingensis* and *L. angelicifolium* should be classified as members of the genus *Melanosciadium*, and suitable taxonomical treatments have been proposed. Meanwhile, a comprehensive description of the new species, *M. jinzhaiensis*, is presented, encompassing its habitat environment and detailed morphological traits.

## 1. Introduction

Apioideae, the largest subfamily of Apiaceae, currently consists of approximately 380 genera and 3200 species [1]. These species are distributed worldwide, with a concentration in northern temperate regions [2]. This subfamily is of great significance as it encompasses a wide range of edible plants containing numerous foods (e.g., *Daucus carota* and *Apium graveolens*), herbs (e.g., *Levisticum officinale* and the species of *Angelica* L.), and spices (e.g., *Coriandrum sativum* and *Foeniculum vulgare*). However, it should be noted that Apioideae also comprises poisonous species (e.g., *Conium maculatum* and *Cicuta viros*). Despite the importance and familiarity of many Apioideae members, uncertainties still exist in the classification of this subfamily due to gaps in our understanding of its phylogenetics.

The most influential and widely adopted classification of Apiaceae was that of Drude (1898) [3], which was later updated and slightly modified by Pimenov and Leonov (1993) [4]. However, the tribes identified in these classifications are primarily based on fruit characteristics, which may not always reflect monophyletic groups. The advent of molecular biology has allowed for the use of an increasing number of molecular markers in phylogenetic analyses of Apioideae. These markers include nuclear ribosomal DNA (nrDNA), internal transcribed spacer (ITS) sequences, and plastid loci and have even revealed that some genera (e.g., *Angelica* L., *Ligusticum* L., *Peucedanum* L., and *Seseli* L.) are not monophyletic [5,6,7,8,9,10,11,12,13,14,15]. More recently, high-throughput DNA sequencing techniques have been employed in Apioideae research, such as the sequencing of the carrot genome [16], transcriptomes [17], whole chloroplasts [18,19,20,21,22,23], and the analysis of the Angiosperms353 gene [24]. However, it is worth noting that most of these studies are broad-scale investigations rather than focused studies on specific taxa.

*Melanosciadium* H. de Boissieu was first published in 1902 and used *M. pimpinelloideum* as the type of species. *Melanosciadium* was treated as a monotypic and endemic genus distributed in the southwest of China and, as such, it was in almost all regional Apiaceae treatments [25,26,27]. *M. pimpinelloideum* is also a traditional Chinese medicine that can be used as folk medicine in Hubei Province, China [28]. Pimenov’s (2006) study of *Melanosciadium* suggested that the genus should contain three species—*M. pimpinelloideum* H. de Boissieu (Figure 1A), *M. bipinnatum* (Shan and F. T. Pu) Pimenov and Kljuykov (Figure 1B), and *M. genuflexum* Pimenov and Kljukov [29]—and this view persisted in the 2017 publication [30]. In addition to Pimenov’s research, only Tan et al. (2015) suggested that *Pimpinella rhomboidea* var. *Tenuiloba* Shan et Pu has been proposed as a synonym for *M. bipinnatum* [31]. In the previous studies on *Angelica* and the related species, it can be found that *Angelica tsinlingensis* K. T. Fu (Figure 1C) is very close to *M. pimpinelloideum* [32,33].

*Angelica tsinlingensis* was first described in 1981 and was classified as a species belonging to the genus *Angelica* L. in both the *Flora Reipublicae Popularis Sinicae* and the *Flora of China* [25,26]. Liao et al. (2013) conducted phylogenetic studies on *Angelica* and its Allies based on the nrDNA and cpDNA sequences [32]. The results indicated a close relationship between *A. tsinlingensis* and *Melanosciadium pimpinelloideum*. However, upon careful examination of the Chinese Apiaceae plant specimens, Pimenov (2017) concluded that *A. tsinlingensis* should be considered a synonym of *Hansenia forbesii* (H. Boissieu) Pimenov and Kljuykov [30]. This determination was made based on Pimenov’s consultation of the aforementioned specimens. After conducting field observations of *A. tsinlingensis*, *M. pimpinelloideum*, and *H. forbesii*, our research prompted us to question the accurate taxonomic classification of *A. tsinlingensis*. For example, in *H. forbesii*, the ultimate segments are oblong-ovate, the rays are nearly equal, the petals are pale yellow or yellowish-green, the dorsal and lateral ribs are winged, the wing width is nearly equal, and the endosperm is concave. In contrast, *A. Tsinlingensis* exhibits distinct characteristics. The median leaflets are rhombic-obovate, and the lateral leaflets are broad-ovate. The rays are extremely unequal, the petals are white, the dorsal ribs are narrow-winged, the lateral ribs are conspicuously wider than the dorsal, and the endosperm plane is slightly concave. Based on these morphological characteristics, we believe that these characteristics are more consistent in the genus *Melanosciadium*.

*Ligusticum angelicifolium* Franch. (Figure 1D) was published in 1894 by Franch. Then, according to the fruit morphology study, Leute considered it should be a species in the genus *Ligusticopsis* Leute, named *Ligusticopsis angelicifolia* (Franch.) Leute [34]. However, Kljuykov argued that it should be the species to which it belongs, and the scientific name of it should be *Angelica angelicifolia* (Franch.) Kljuykov and Pimenov have identified this species as a member of the *Angelica* in their respective studies [30,35]. In addition, Lavrova has also suggested that *Ligusticum angelicifolium* may be a synonym of *Conioselinum angelicifolium* (Franch.) Lavrova [36]. Currently, there are two predominant perspectives regarding the nomenclature of this species. One viewpoint advocates for the name *Ligusticopsis angelicifolia*, while the other argues that it should be referred to as *A. angelicifolia*. So, after field surveys, consulting specimens, and performing DNA sequencing, we analyzed the location of this species.

During our field trips in August 2018, we collected specimens belonging to the Apiaceae family. These specimens were found to be flourishing at elevations ranging from 700 to 1100 m above sea level in Jinzhai County, located in the southwest region of Anhui Province, China. The specimen is similar to *Melanosciadium pimpinelloideum*. After carefully observing morphological characters and molecular analyses and checking the specimens of *Melanosciadium* and *A. cincta* at the online herbariums, we consulted with the relevant floras and literature [25,26]. Based on the comprehensive analysis of all available evidence, it is our contention that the observed species can be classified as a novel species belonging to the genus *Melanosciadium*. Therefore, the present study provides a description and illustration of *M. jinzhaiensis* (Figure 2). 

The plastid genome has many features, such as monolepsis, small subfractions, multiple replications, and moderate nucleotide substitution rates [37,38,39,40,41]. Despite the highly variable characteristics of the plastid genome, it has the potential to obtain a robust phylogenetic tree [19,21,42,43,44,45]. With the advancement of next-generation sequencing and bioinformatics technologies, obtaining plastid genome data is now more affordable and faster than ever before. As a result, plastid genome sequences have been widely and successfully utilized in plant phylogenetic analyses [46,47,48,49,50]. 

In this study, we sequenced and assembled the plastid genome sequences of *Melanosciadium pimpinelloideum*, *M. bipinnatum*, *Angelica tsinlingensis*, *Ligusticum angelicifolium*, the new species *M. jinzhaiensis*, and another three *Angelica* species. Then, together with published plastid genome sequences from the family Apiaceae, we reconstructed the phylogenetic position of *A. tsinlingensis*, *L. angelicifolium*, and the new species *M. jinzhaiensis* based on nrDNA sequences and plastid genome sequences. We also performed comparative plastid genome analyses between these species. We provide a taxonomic revision for *A. tsinlingensis*, *L. angelicifolium*, and the new species *M. jinzhaiensis* based on plastid genome comparison and phylogenomic and morphological evidence.

## 2. Material and Methods

### 2.1. Taxon Sampling

We have collected *Melanosciadium pimpinelloideum*, *M. bipinnatum*, *Angelica tsinlingensis*, *Ligusticum angelicifolium*, *M. jinzhaiensis*, and related species. Specimen vouchers are stored in the Herbarium of Sichuan University (SZ; Chengdu, China). The rest of the species sequences for our phylogenetic analysis were obtained from the National Center for Biotechnology Information (NCBI). 

### 2.2. Morphological Study

Fruits from the wild specimens of *Melanosciadium pimpinelloideum*, *Angelica tsinlingensis*, *Ligusticum angelicifolium*, and *M. jinzhaiensis* were collected for morphological study. Morphological analyses of other characters were based on plants in the wild and herbarium specimens. The anatomical study of the fruit was carried out through a hand section for observation of the details (Figure 3 and Figure 4). No less than 10 seeds were observed for each species. The specimen vouchers are stored in the Herbarium of Sichuan University (SZ), and the corresponding details can be found in Appendix A.

### 2.3. nrDNA Sequences and Plastid Genome Sequencing, Assembly, and Annotation

The DNA sequencing was performed using the Illumina Novaseq 6000 platform (Illumina, San Diego, CA, USA) at Novogene (Beijing, China) with the Novaseq 150 sequencing strategy. The clean data that remained after processing were assembled using NOVOPlasty 2.7.1 [51] with the default K-mer value of 39. The rbcL sequence of *Melanosciadium pimpinelloideum* (GenBank accession No.: KX527530.1) was used as the initial seed input, and plastid genome sequences of *M. pimpinelloideum* (GenBank accession No.: MW436383) was used as the reference for these species. Preliminary genome annotation was performed utilizing the PGA program [52]. Subsequently, manual adjustments were made to uncertain genes, as well as uncertain start and stop codons, by comparing them with other related plastid genomes using Geneious R11 [53]. Protein-coding sequences (CDS) extraction was conducted using the PhyloSuite v1.2.3 software [54]. The annotated genome of the species was submitted to the National Center for Biotechnology Information (NCBI), and the accession number was listed in Appendix A. Additionally, in order to enhance the reliability of the phylogenetic data, we employed the program GetOrganelle v1.7.5 [55] to extract the nrDNA sequences, which consists of ETS, 18S rRNA, ITS 1, 5.8S rRNA, ITS 2, complete sequence; and 26S rRNA partial sequence. The relevant information and accession number for the National Center for Biotechnology Information (NCBI) of nrDNA sequences are listed in Appendix A.

### 2.4. Plastid Genome Comparative Analyses

The sequenced species’ annotated genome sequences were submitted to GenBank, and their corresponding accession numbers are listed in Appendix A. Circular gene maps of the annotated genomes were performed using the online program Chloroplot (https://irscope.shinyapps.io/chloroplot/; accessed on 20 December 2023) [56].

The junctions between single-copy regions (LSC region and SSC region) and inverted repeat regions (IRA and IRB regions) among *Melanosciadium pimpinelloideum*, *Angelica tsinlingensis*, *Ligusticum angelicifolium*, *M. jinzhaiensis* and another three *Angelica* species (*A. dahurica*, *A. cincta*, and *A. sylvestris*; total 7 species) were compared by using Geneious R11 [53], then visualized it by manually.

### 2.5. Phylogenetic Analysis

In this study, we used nrDNA sequences and cpCDS sequences. Our phylogenetic analyses utilized data from the National Center for Biotechnology Information (NCBI). The data of nrDNA sequences (includes ETS, 18S rRNA, ITS 1, 5.8S rRNA, ITS 2, complete sequence; and 26S rRNA partial sequence) were listed in Appendix A, and the data of plastid genome sequences were listed in Appendix A. 

We used MEGA7 [57] to align DNA sequences with manual adjustment to make sequences more aligned, positioning the gap to minimize nucleotide mismatch. Phylogenetic analyses were carried out employing Maximum Likelihood (ML) and Bayesian Inference (BI) analyses. Before starting, the program MrModeltest version 2.2 [58] was used to select the best model of the ITS sequences and CDS sequences nucleotide substitution. The best model of data I and data II were GTR + G + I for both ML analyses and BI analyses, and the best model of data III was GTR + G for ML analyses and GTR + G + I for BI analyses. Maximum likelihood (ML) analyses were undertaken using RAxML v8.2.4 [59] with 1000 bootstrap replicates. The bootstrap value (BS) is closer to 100%, the more accurate it is. Bayesian Inference (BI) analyses were performed by MrBayes version 3.2 [60]. Four simultaneous runs were performed using Markov chain Monte Carlo (MCMC) simulations for 10 million generations. The analysis began with a random tree and sampled one tree every 1000 generations. The first 20% of obtained trees were discarded as “burn-in”, and the remaining were used to calculate a majority-rule consensus topology and posterior probability (PP) values. Based on previous analyses, the species *Chamaesium* H. Wolff and *Sanicula* L. were chosen as the outgroup [17,61]. 

## 3. Result and Discussion

### 3.1. Morphological Result

Following a field observation, the fruit morphological characters were investigated and compared to the related species. The results of the morphological study indicate that *Melanosciadium jinzhaiensis* is similar to *M. pimpinelloideum* rather than *Angelica biserrata*, though the new species’ specimens were previously identified as *A. biserrata* (KUN1485850!, NAS00038853!). In our study, we found that the morphological characteristics of *M. jinzhaiensis* bear a closer resemblance to those of *M. pimpinelloideum* (Figure 1A and Figure 2; Figure 3A and Figure 4). These similarities encompass leaf forms, leaflets, sheaths, and umbel rays. They share some common characteristics, such as stems purplish, sheaths margin purplish, leaflets rhombic-obovate, bracts often absent, rays very un-equal, etc.

For *Angelica tsinlingensis* (Figure 1C and Figure 3C), our morphological study reveals that its rays are short and very un-equal, leaflets rhombic-obovate, bracts absent, vittaes abundant, and endosperm concave. We think these characteristics bear a closer resemblance to the attributes exhibited by *Melanosciadium*. The situation of *Ligusticum angelicifolium* (Figure 1D and Figure 3D) displayed resemblances to that of *A. tsinlingensis* and, in fact, was even more pronounced. Pimenov believed that *A. Tsinlingensis* was the synonym of *Hansenia forbesii* [30,62]. In our field investigation, we found that *H. forbesii* and *A. Tsinlingensis* had obvious morphological differences. For example, in *H. forbesii*, the ultimate segments are oblong-ovate, the rays are nearly equal, the petals are pale yellow or yellowish-green, the dorsal and lateral ribs are winged, the wing width is nearly equal, and the endosperm is concave [25,26,62]. However, *A. Tsinlingensis* exhibits distinct characteristics. The median leaflets are rhombic-obovate, and the lateral leaflets are broad-ovate. The rays are extremely unequal, the petals are white, the dorsal ribs are narrow-winged, the lateral ribs are conspicuously wider than the dorsal, and the endosperm plane is slightly concave [25,26]. These features can be easily distinguished from *H. forbesii*. According to previous studies and molecular phylogenetic results, we can find that *H. forbesii* is located in the East Asia Clade, whereas *A. Tsinlingensis* is located in Selineae [13,14,32,63,64]. Consequently, *A. Tsinlingensis* is a separate species and should belong to the genus *Melanosciadium*.

The overall morphology of *Ligusticum angelicifolium* was very similar to that of *M. bipinnatum*, such as leaf forms, leaflets, umbel rays, and ray number [25,26,31]. And the fruit morphology of *L. angelicifolium* is more similar to that of *A. tsinlingensis* and *M. jinzhaiensis* (Figure 3 and Figure 4). These species can be easily distinguished by mericarps characters, and the main morphological differences of these species are summarized in Table 1.

### 3.2. Plastid Genome Comparation Results

These total seven species’ plastid genomes presented a single and typical quadripartite circular structure (Figure 5) that was divided into four regions: two inverted repeat regions (IRs), a large single-copy region (LSC), and a small single-copy region (SSC). The size of the plastid genomes of three *Angelica* species (*A. dahurica*, *A. cincta*, and *A. sylvestris*) ranged from 146,138 bp (*A. sylvestris*) to 146,847 bp (*A. dahurica*). And according to a previous study [49,65], the size of the plastid genomes of *Angelica* is around 147,000 bp. However, the size of the plastid genomes in *Melanosciadium pimpinelloideum*, *A. tsinlingensis*, *Ligusticum angelicifolium*, and *M. jinzhaiensis* are as follows: *A. Tsinlingensis* is 163,608 bp, *L. angelicifolium* is 163,798 bp, *M. pimpinelloideum* is 164,329 bp, and *M. jinzhaiensis* is 164,544 bp, respectively. These four species are very similar in size. Large differences in genome size led to differences in the number of genes. In three *Angelica* species (*A. dahurica*, *A. cincta*, and *A. sylvestris*), including a total of one hundred and twenty-nine genes (eighty-four protein-coding genes, thirty-six transfer RNA genes, eight ribosomal RNA genes, and one pseudogene), while the remaining four species (*M. pimpinelloideum*, *A. tsinlingensis*, *Ligusticum angelicifolium*, and *M. jinzhaiensis*) including a total of one hundred and forty-four genes (ninety-eight protein-coding genes, thirty-seven transfer RNA genes, eight ribosomal RNA genes, and one pseudogene). They shared one hundred and fourteen unique genes, including eighty protein-coding genes, thirty transfer RNA genes, four ribosomal RNA genes, and one pseudogene (Table 2). 

The length of small single-copy regions (SSC) of all these seven species is consistent; this result is similar to previous plastid genome research on Apioideae (Apiacea) [18,19,20,21,49,66,67]. In contrast to the SSC regions, the large single-copy regions (LSC) showed significant variation. In three *Angelica* species (*A. dahurica*, *A. cincta*, and *A. sylvestris*), the LSC regions ranged from 93,297 bp (*A. cincta*) to 93,539 bp (*A. dahurica*). The length of the LSC regions for the other four species is 76,649 bp (*A. tsinlingensis*), 76,900 bp (*Ligusticum angelicifolium*), 76,578 bp (*Melanosciadium jinzhaiensis*), and 76,450 bp (*M. pimpinelloideum*). This is caused by the expansion of the IR regions. The expansion and contraction of IR regions are important for variations in genome size and play a crucial role in plant evolution [18,61,68,69,70]. 

In our study, the IR regions ranged from 17,817–18,058 bp in the three *Angelica* species (*A. dahurica*, *A. cincta*, and *A. sylvestris*), while the length of IR regions in these four species is 34,705 bp (*A. tsinlingensis*), 34,717 bp (*Ligusticum angelicifolium*), 35,212 and 35,215 bp (*Melanosciadium jinzhaiensis*), and 34,717 bp (*M. pimpinelloideum*), respectively. Meanwhile, the junctions of the IR/LSC have changed, and the junctions of the IR/SSC are consistent. We illustrated the junctions of IR/LSC and IR/SSC and designated J_LA_ (LSC/IRA), J_LB_ (LSC/IRB), J_SA_ (SSC/IRA), and J_SB_ (SSC/IRB) (Figure 6). Junctions J_SB_ and J_SA_ are consistently positioned across all species: J_SB_ is in between the pseudogene *ψycf1* and the *ndhF* gene, and J_SA_ occurs in the *ycf1* gene. In three *Angelica* species (*A. dahurica*, *A. cincta*, and *A. sylvestris*), junction J_LB_ occurs in the *ycf2* gene, and junction J_LA_ between the *trnL-CAA* gene and *trnH-GUG* gene. But in the other four species (*M. pimpinelloideum*, *A. tsinlingensis*, *Ligusticum angelicifolium*, and *M. jinzhaiensis*), junction J_LB_ occurs in the *petB* gene and junction J_LA_ between the *petD* gene and *trnH-GUG* gene. 

In conclusion, we observed similarities in the plastid genome size, gene number, and length of the LSC and IR regions among the four species: *M. pimpinelloideum*, *A. tsinlingensis*, *Ligusticum angelicifolium*, and *M. jinzhaiensis*. The plastid genomes of these species are distinct from those of the three *Angelica* species (*A. dahurica*, *A. cincta*, and *A. sylvestris*).

### 3.3. Phylogenetic Result

For nrDNA sequences, the phylogenetic results are presented in Figure 7. The topological consistency of the phylogenetic tree is derived from Bayesian Inference (BI) and Maximum Likelihood (ML) analyses. Only the BI tree is depicted in Figure 7, accompanied by bootstrap support values derived from ML analyses. The phylogenetic exhibited that *Melanosciadium jinzhaiensis*, *M. pimpinelloideum*, *M. bipinnatum*, *Angelica tsinlingensis*, and *Ligusticum angelicifolium* formed a monophyletic group with robust support (Bayesian inference posterior probability, BI = 0.99; maximum parsimony bootstrap, ML = 67%). This monophyletic group is sister to *Angelica* and is located in the Selineae [13,14,32,71].

Our cpCDS sequences of plastid genome phylogenetic analyses used the NCBI data. The data are listed in Appendix A. The phylogenetic trees derived from BI and ML analyses were topologically consistent. Thus, only the BI tree is shown in Figure 8, with bootstrap support values obtained from ML analyses. The phylogenetic tree of plastid genomes also showed these five species (*Melanosciadium jinzhaiensis*, *M. bipinnatum*, *M. pimpinelloideum*, *Angelica tsinlingensis*, and *Ligusticum angelicifolium*) formed a monophyletic group with the best support (Bayesian inference posterior probability, BI = 1.00; maximum parsimony bootstrap, ML = 100%). The monophyletic group is also a sister to *Angelica* within the Selineae clade [13,14,32,71].

Therefore, based on morphological evidence, plastid genome comparison results, and phylogenetic results based on both nrDNA sequences and cpCDS sequence, we contend that *Angelica tsinlingensis* and *Ligusticum angelicifolium* should be classified as members of the genus *Melanosciadium*. Additionally, we support the classification of *Melanosciadium jinzhaiensis* (Figure 9) as a new species within the genus *Melanosciadium*, Apiaceae.

## 4. Taxonomic Treatment

(1)*Melanosciadium jinzhaiensis* Q. P. Jiang and X. J. He *sp.nov.* (Figure 2, Figure 4, Figure 9 and Appendix A)

Type: China. Anhui Province: Jinzhai County, elevation 700–1300 m a.s.l., Q. P. Jiang, JQP18082901, JQP180102302, flowering and fruiting (holotype SZ!, isotype SZ!).

Diagnosis: Sheath margin is purplish, glabrous, and not inflated; leaf blade is broadly ovate-triangular; and median leaflets are short-petiolulate, rhombic-obovate. Rays 3–12 are 0.3–2.5 cm, very un-equal, and hispidulous; bracteoles 3–5 are linear, 2–6 mm, and unequal. Petals are broad-obovate, purplish, and apex incurved. Mericarps have 3–5 ribs, dorsal ribs that are narrow-winged to winged, lateral ribs that are broad-winged, Vittae are 2–3 (4) in each furrow, often 6 (or more) on commissure.

Description

Plants are perennial, 40–140 cm. They are root conic. Stem is thinly ribbed, glabrous to hispidulous upward, and branched; branches of stems are purplish. Basal and lower petioles are long; petioles shorten upward; sheathing is at base, sheath margin is purplish, glabrous; leaf blade is broadly ovate-triangular, 15–30 × 10–25 cm, 1–2-ternate; median leaflets are short-petiolulate, rhombic-obovate, 10–15 × 5–10 cm, base cuneate; lateral leaflets are broad-ovate, sometimes 1–2-lobed, base truncate, margin incised-serrate, teeth mucronate, and apex acuminate or cuspidatus. Umbels are 1–5 cm across; peduncles are 3–16 cm, hispidulous; bracts are often absent, rays 3–10 are 0.3–2.5 cm, very un-equal, hispidulous; bracteoles 3–5 are linear, 2–6 mm, unequal; umbellules are 5–17-flowered. The calyx teeth are obsolete or minute broad-triangular-ovate. Petals are broad-obovate, purplish, apex incurved. Stylopodium is low-conic; its styles are long, beyond petals, fruit oval or ellipsoid, 2.4–4.1 × 4–7 mm, dorsally compressed. Mericarps have 3–5 ribs, dorsal ribs are narrow-winged to winged, lateral ribs are broad-winged, and endosperm plane is slightly concave. Vittae 2–3 (4) are in each furrow, often 6 (or more) on commissure. Fl. July–September, fr. August–October. 

Etymology

The specific epithet refers to the type locality, Jinzhai County of Anhui Province in China.

Phenology

Flowering from July to September and fruiting from August to October.

Distribution, habitat, and ecology 

This species is currently known little from the Herbariums, only located in the type locality, Jinzhai County of Anhui Province in China. According to its natural environment, we speculate that it may grow in the forests of eastern Anhui, western Hubei, and northern Jiangxi at an altitude of 700–1500 m. This species grows in the humid environment under the forests. Endemic to China.

Additional specimens examined (paratypes)

China: Anhui Province: Lu’an city, Jinzhai county, Tiantangzhai town, 20 October 2019, 713 m, Xin-Xin Zhu, Jun Wang, Shu Lin, Dong-Sheng Fang, ZXX191565 (KUN!); Jinzhai county, Baimazhai Forestry Farm of Jinzhai county, 23 October 1984, 1300 m, Mao-Bin Deng, Hong-Tu Wei, 82,025 (NAS!). 

(2)*Melanosciadium tsinlingensis* (K. T. Fu) Q. P. Jiang and X. J. He comb. nov.

≡ *Angelica tsinlingensis* Fu Kuntsun, 1981, Fl. Tsinling. 1(3): 420,461, Figures 358. 

Type: CHINA. “Shenxi, Hwa-in Hsien, Hwashan, Ta-pai-yang-cha, alt. 1500 m, 22. 09. 1974, Fu Kuntsun 17,242” (holotype WUK!).

Diagnosis: Blade 2-ternate, median leaflets are often short-petiolulate, rhombic-obovate; lateral leaflets are broad-ovate, often 1–2 lobed; margin is incised-serrate, apex acuminate. Rays 9–20 are very unequal, bracts are absent. Petals are broad-obovate, white; the outer is slightly enlarged. Fruit is oblong to suborbicular, dorsal ribs are narrow-winged, lateral ribs are conspicuously wider than the dorsal; endosperm (at commissural side) plane is concave, vittae 2–4 are in each furrow, 4–8 on commissure.

Phenology

Flowering occurs from August to September, and fruiting occurs from September to October.

Distribution, habitat, and ecology

Distribution is found in the Gansu Province and Shaanxi Province. They grow in the forests or shrubby thickets at an altitude of 1200–2300 m. They are endemic to China.

Additional specimens examined

China: Gansu Province: Tianshui, 26 July 1951, 1920 m, Zhen-Wan Zhang, 152 (WUK!); Tianshui, 10 August 1951, 1600 m, Ji-Meng Liu 10,406 (WUK!); Tianshui, 18 July 1964, Kun-Jun Fu, 16,044 (WUK!).

China: Shaanxi Province: Huayin City, Mt. Hua, 2 September 1956, 1250 m, Kun-Jun Fu, 8548 (NAS!); Mei County, Mt. Taibai, 23 September 1988, 2100 m, Zhenhai Wu, 88–718 (PE!); Huayin City, Mt. Hua, 13 August 1966, 2030 m, Zuo-Bing Wang, 19,696 (KUN!,WUK!); Huayin City, Mt. Hua, 5 September 1956, Kun-Jun Fu, 8558 (KUN!); Huayin City, Mt. Hua, 19 September 1973, Zuo-Bing Wang, 40 (WUK!); Huayin City, Mt. Hua, 23 September 1974, 1700 m, Kun-Jun Fu, 17,263 (WUK! 0416082,); Mei County, 9 October 1973, 1575 m, Kun-Jun Fu, 17,150 (WUK!); Mei County, 11 October 1973, 1700 m, Kun-Jun Fu, 17,183 (WUK!); Mei County, 11 October 1973, 1700 m, Kun-Jun Fu, 17,182 (WUK!); Mei County, 10 October 1973, 1600 m, Kun-Jun Fu, 17,157 (WUK!); Mei County, 10 October 1973, 1600 m, Kun-Jun Fu, 17,154 (WUK!); Mei County, 10 October 1973, 1750 m, Kun-Jun Fu, 17,165 (WUK!); Mei County, 10 October 1973, 1800 m, Kun-Jun Fu, 17,168 (WUK!); Mei County, 10 October 1973, 1700 m, Kun-Jun Fu, 17,162 (WUK!); Mei County, 10 October 1973, 1720 m, Kun-Jun Fu, 17,164 (WUK!); Mei County, 10 October 1973, 1880 m, Kun-Jun Fu, 17,178 (WUK!); Mei County, 10 October 1973, 1850 m, Kun-Jun Fu, 17,173 (WUK!); Huayin City, 22 September 1974, 1700 m, Kun-Jun Fu, 17,244 (WUK!); Huayin City, Mt. Hua, 20 September 1973, 1100 m, Hong-Jie Wang, 54 (WUK!); Huayin City, Mt. Hua, 19 September 1973, 2000 m, Hong-Jie Wang, 31 (WUK); Huayin City, Mt. Hua, 19 September 1973, 2000 m, Hong-Jie Wang, 28 (WUK!); Huayin City, Mt. Hua, 19 September 1973, 2000 m, Hong-Jie Wang, 30 (WUK!); Mt. Qingfeng, August 1940, Shi-Bian Zhao, 140 (WUK!); Mei County, Mt. Taibai, August 1940, Zhen-Hua Wang, 1050 (WUK!); Mei County, 10 October 1973, 1780 m, Kun-Jun Fu, 17,166 (WUK!); Huayin City, 23 September 1974, 1700 m, Kun-Jun Fu, 17,263 (WUK!); Hu County, 06 September 1962, 1600 m, Kun-Jun Fu, 14,246 (WUK!); Huayin City, 28 June 1974, 1700 m, Kun-Jun Fu, 17,228 (WUK!); Huayin City, 3 August 1973, 1300 m, Kun-Jun Fu, 16,870 (WUK!); Huayin City, 5 August 1973, 1900 m, Kun-Jun Fu, 16,892 (WUK!); Huayin City, 22 September 1974, 1500 m, Kun-Jun Fu, 17,242 (WUK!); Mei County, 10 October 1973, 1790 m, Kun-Jun Fu, 17,167 (WUK!); Huayin City, 19 September 1973, 1255 m, H.J.Wang 9 (WUK); Huayin City, 19 September 1973, 2050 m, Hong-Jie Wang, 40 (WUK!); Huayin City, 19 September 1973, 1285 m, Hong-Jie Wang, 9 (WUK!); Huayin City, 19 September 1973, 1785 m, Hong-Jie Wang, 80 (WUK!); Huayin City, Mt. Hua, 2 September 1956, 1250 m, Kun-Jun Fu, 8548 (WUK!); Huayin City, 27 May 1938, 1500 m, W.Y.Hsia, 4460 (WUK!).

(3)*Melanosciadium angelicifolium* (Franch.) Q. P. Jiang and X. J. He comb. nov.

≡ *Angelica angelicifolia* (Franch.) Kljuykov, 1999, Bot. Zhurn. (St. Perersburg) 84(3): 482, Figure 1A,B.

≡ *Ligusticum angelicifolium* Franch., 1894, Bull. Annuel Soc. Philom. Paris (sér. 8) 6: 133.

≡ *Ligusticopsis angelicifolia* (Franch.) Leute, 1969, Ann. Naturhist. Mus. Wien 73: 70, Table 3, Figure 2.

≡ *Conioselinum angelicifolium* (Franch.) Lavrova, 2002, Abstr. Intern. Sci. Conf. Syst. Higher Pl. (Moscow): 67.

Type: CHINA. “Yun-nan, les bois de San tcha ho au-dessus de Mo-so-yn, à 3000 m d’altit., 05 IX 1887, Delavay 2970” (lectotype P! (barcode P03224578), designated by G.-H. Leute, 1969: 70; isolectotypes NY (barcode NY00406024); P! (barcodes P03224574, P03237451, P03237452, P03237453, P03237464, P03237466, P03237469)); “ibid., Delavay 3942, 4093” (syntypes A (barcode A00112218); GH (barcode GH00112219); K! (barcode K000685240); P! (barcodes P03237455, P03237456, P03237458, P03237467)); “in umbrosis prope Fang-yang-tchang, 14 X 1887, Delavay 3707” (syntypes P! (barcodes P03224579, P03237454, P03237457)).

Diagnosis: Stem and sheaths green or purplish or purple. Blade is 2–3-ternate, and the ultimate segments are rhombic-obovate to oblong-ovate, with margins serrate. Rays 18–25 are very unequal. Petals are white to purple and anther is white to purple. The fruit is oblong-ovoid, with dorsal ribs narrow-winged, lateral ribs broad-winged, and endosperm slightly concave.

Phenology

Flowering occurred from July to August, and fruiting occurred from August to September.

Distribution, habitat, and ecology

In Yunnan Province, they grow in the shade of wet forest, forest edge grassland, or hillside grass at an altitude of 2500–4000 m. They are endemic to China.

Additional specimens examined

China: Yunnan Province: Lijiang City, Lijiang Snow Mountain, 25 September 1940, Ren-Chang Qin, 31,070 (PE!, KUN!); Lijiang City, 4 July 1939, R.C.Ching 30,350 (PE!); Lijiang City, Lijiang Snow Mountain, 29 July 1940, Ren-Chang Qin, 30,911 (PE!, KUN!); Lijiang City, 4 July 1939, Xi-Guang Zhao, 30,350 (KUN!); Heqing City, 23 August 1929, 2900 m, Ren-Chang Qin, 24,021 (KUN!); Lijiang City, 2 September 1958, 2900 m, Wen-Cai Wang, 115 (KUN!); Lijiang City, 06 August 1959, 3200 m, 22,537 (KUN!); Heqing City, 8 August 1929, Ren-Chang Qin, 23,585 (KUN!); Chenkang snow range, 3 August 1938, 3500 m, De-Jun Yu, 17,159 (KUN!); Heqing City, August 1963, 1888 (KUN!); Lijiang City, 29 September 1984, Ze-Hui Pan, Chang-Qi Yuan et al., 233 (NAS!); Lijiang City, 29 September 1984, Ze-Hui Pan, Chang-Qi Yuan et al., 202 (NAS!); Lijiang City, 2 August 1986, 3800 m, Bing-Guang He, 81 (NAS!); Lijiang City, 2 August 1986, 3350 m, Bing-Guang He, 85 (NAS!); Xianggelila County, 30 August 2011, 2985 m, Zhen-Dong Fang, Xian Hai, Ai-Fang Xie, Lei Li, Jian-Sheng Wei, DJDC-685 (SABG!).


**Key to determination of *Melanosciadium* species:**


1a.Fruits are oblong-ovoid to subglobose, and all ribs are prominent, ridged.
2a.Petals purplish to purple; rays short, very unequal, rays about 5–14; anthers purple . . . . . . . . . . . . . . . . . . . . . . . . . . . . . . . . . . . . . . . . . . . . . . . . . . . . . . *M. pimpinelloideum*2b.Petals purplish; rays long, very unequal, rays about 14–30; anthers white. . . . . . . . . . . . . . . . . . . . . . . . . . . . . . . . . . . . . . . . . . . . . . . . . . . . . . . . . . . . . . . . *M. bipinnatum*
1b.Fruits oblong-ovoid, oblong or suborbicular, dorsal ribs narrow-winged, lateral ribs winged.
3a.Rays fewer, less than 10.
4a.The rhachis of leaf blades refracted (bent back); rays unequal or subequal. . . . . . . . . . . . . . . . . . . . . . . . . . . . . . . . . . . . . . . . . . . . . . . . . . . . . . . . . . .*M. genuflexum*4b.The rhachis of leaf blades not refracted; ray very unequal and short . . . . . . . . . . . . . . . . . . . . . . . . . . . . . . . . . . . . . . . . . . . . . . . . . . . . . . . . . *M. jinzhaiensis*
3b.Rays more, more than 10.
5a.Rays 10–20; petals white, the outer slightly enlarged . . . . . . *M. tsinlingensis*5b.Rays 10–30; petals white to purple, the outer not enlarged . . . . . . . . . . . . . . . . . . . . . . . . . . . . . . . . . . . . . . . . . . . . . . . . . . . . . . . . . . . . . . . . .*M. angelicifolium*



## Figures and Tables

**Figure 1 plants-13-00907-f001:**
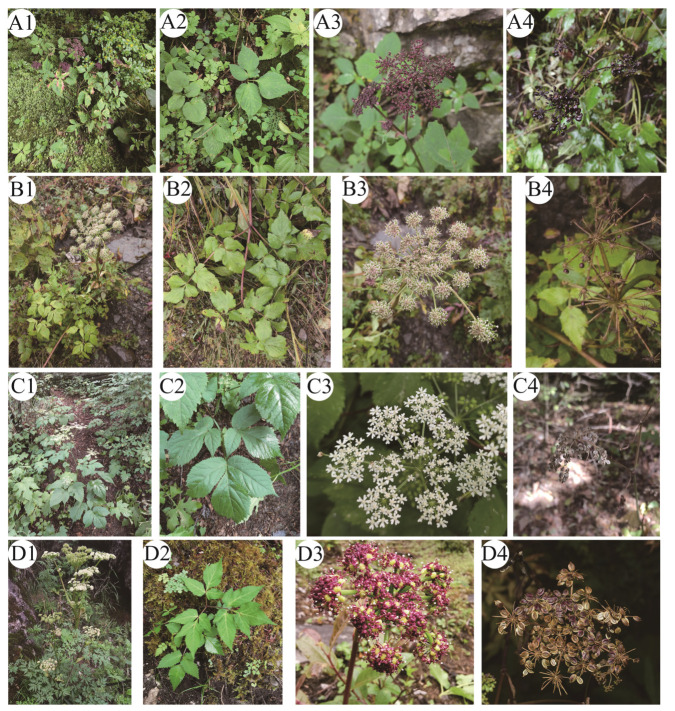
Habitat and umbel rays of the following: (**A**) *M. pimpinelloideum*; (**B**) *M. bipinnatum*; (**C**) *A. tsinlingensis*; (**D**) *L. angelicifolium*. (**A1**,**B1**,**C1**,**D1**) Habitat. (**A2**,**B2**,**C2**,**D2**) Leaf (basal). (**A3**,**B3**,**C3**,**D3**) Flowers. (**A4**,**B4**,**C4**,**D4**) Umbel rays and fruits.

**Figure 2 plants-13-00907-f002:**
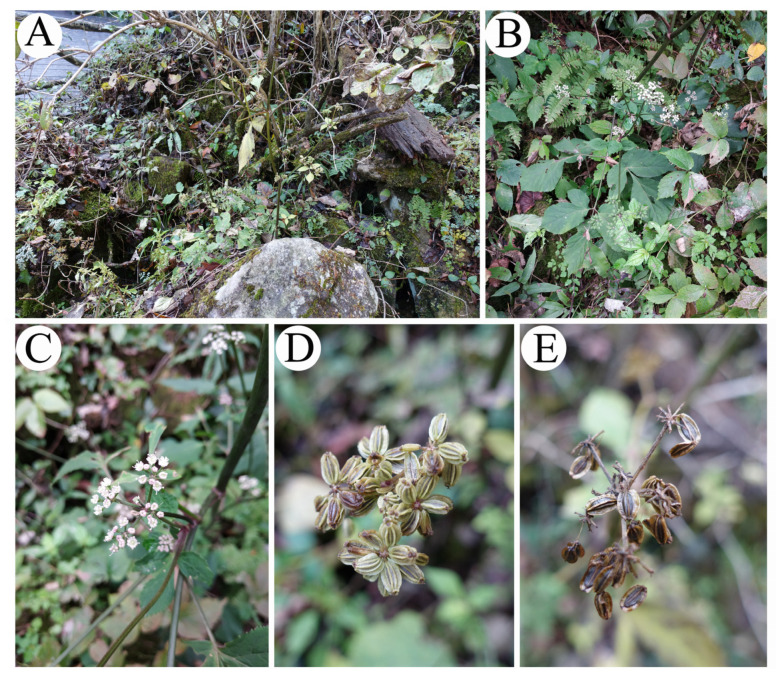
*M. jinzhaiensis* in the field: (**A**) Habitat; (**B**) Basal leaves; (**C**) Flowers; (**D**,**E**) Umbels and fruits.

**Figure 3 plants-13-00907-f003:**
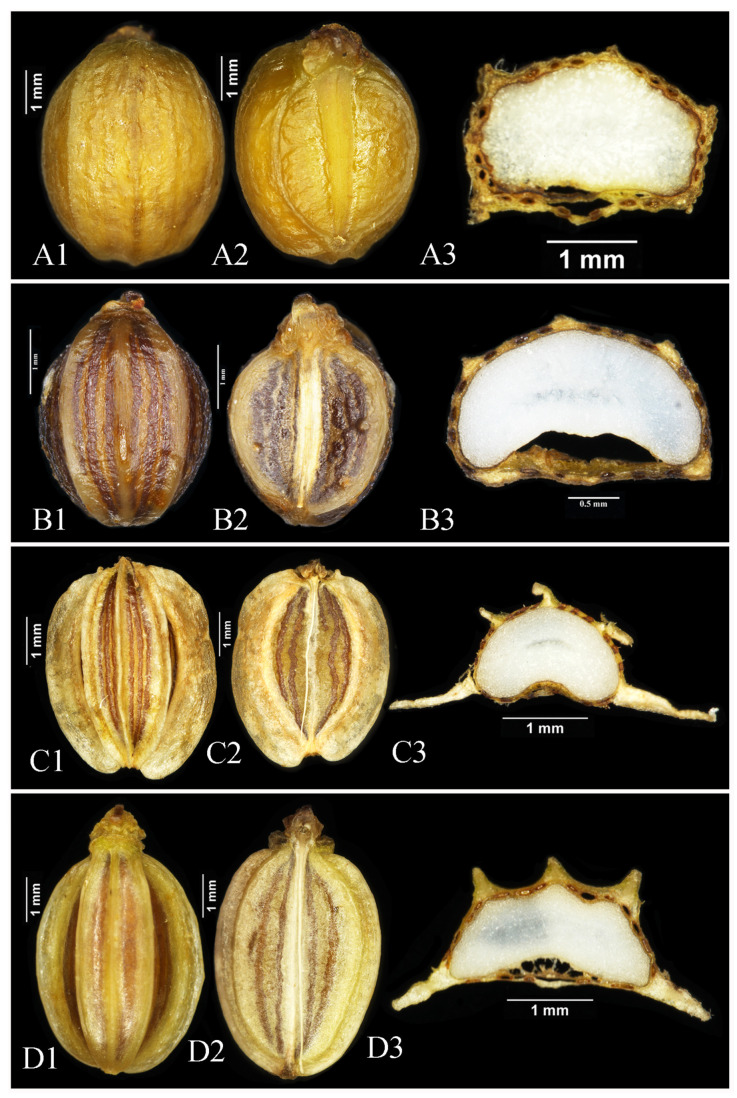
Fruit characters of the following: (**A**) *M. pimpinelloideum*; (**B**) *M. bipinnatum*; (**C**) *A. tsinlingensis*; (**D**) *L. angelicifolium*. (**A1**,**B1**,**C1**,**D1**) Dorsal view of fruits. (**A2**,**B2**,**C2**,**D2**) Commissural side of fruits. (**A3**,**B3**,**C3**,**D3**) Cross-section of fruits (hand section).

**Figure 4 plants-13-00907-f004:**
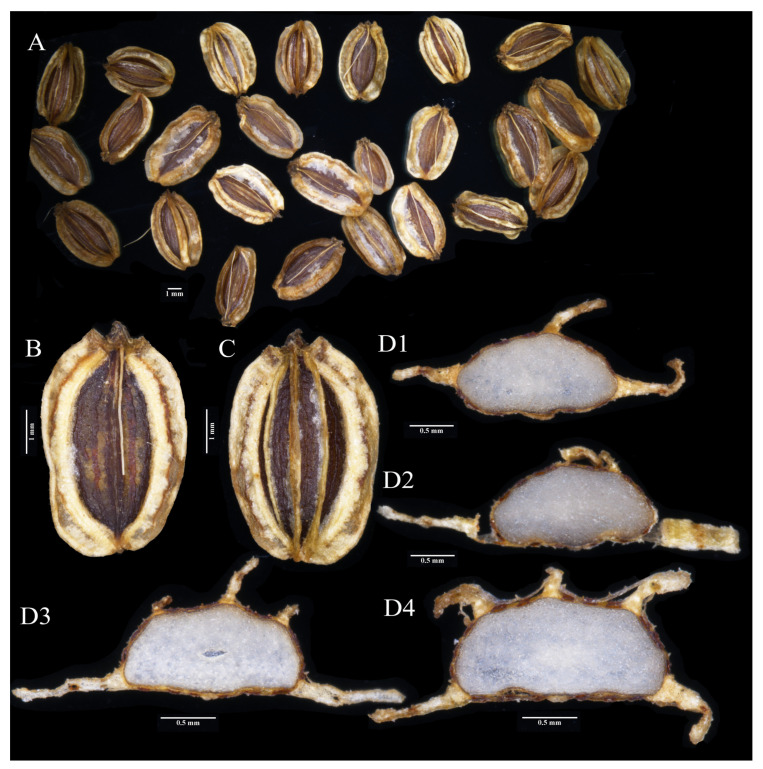
Fruit characters of *Melanosciadium jinzhaiensis*: (**A**) Fruits; (**B**) Commissural side of fruit; (**C**) Dorsal view of fruit; (**D**) Cross-section of fruit (hand section), (**D1**–**D4**), dorsal ribs of varying number and width.

**Figure 5 plants-13-00907-f005:**
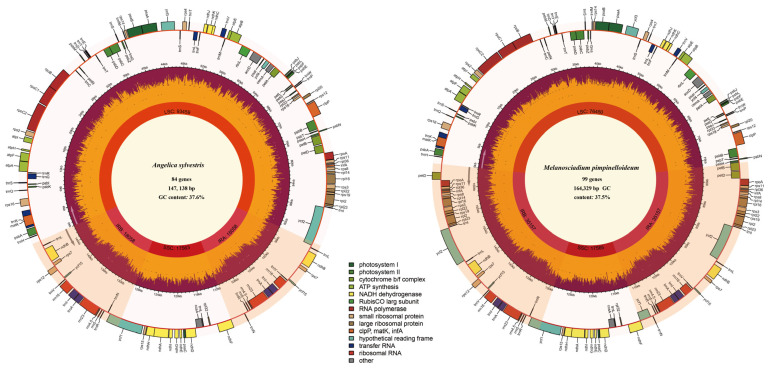
Chloroplast genome map of *Angelica sylvestris* (the **left**) and *Melanosciadium pimpinelloideum* (the **right**). The species name and specific information regarding the genome (length, GC content, and the number of genes) are depicted in the center of the plot. The length of the corresponding single short copy (SSC), inverted repeat (IRa and IRb), and large single-copy (LSC) regions is also given. Genes are color-coded by their functional classification. Represented with arrows, the transcription directions for the inner and outer genes are listed clockwise and anticlockwise, respectively. The optional shaded area stretching from the inner sphere toward the outer circle marks the IR regions.

**Figure 6 plants-13-00907-f006:**
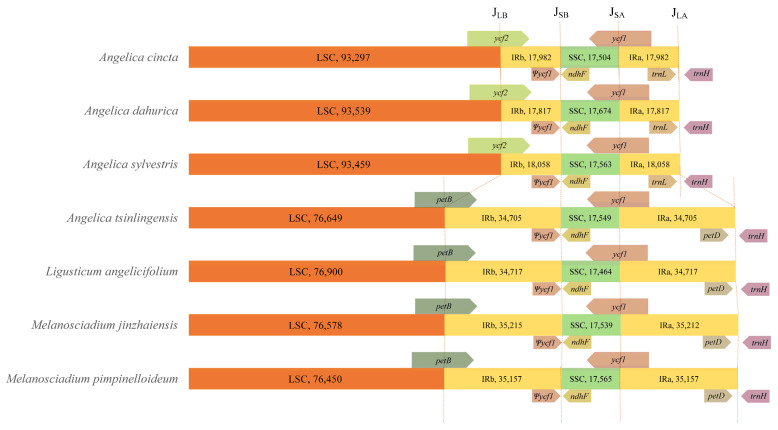
Comparison of the border regions of *Melanosciadium pimpinelloideum*, *Angelica tsinlingensis*, *Ligusticum angelicifolium*, *M. jinzhaiensis,* and another three *Angelica* species (total 7 species) plastid genomes. LSC (large single-copy), SSC (small single-copy), and IR (inverted repeat) regions.

**Figure 7 plants-13-00907-f007:**
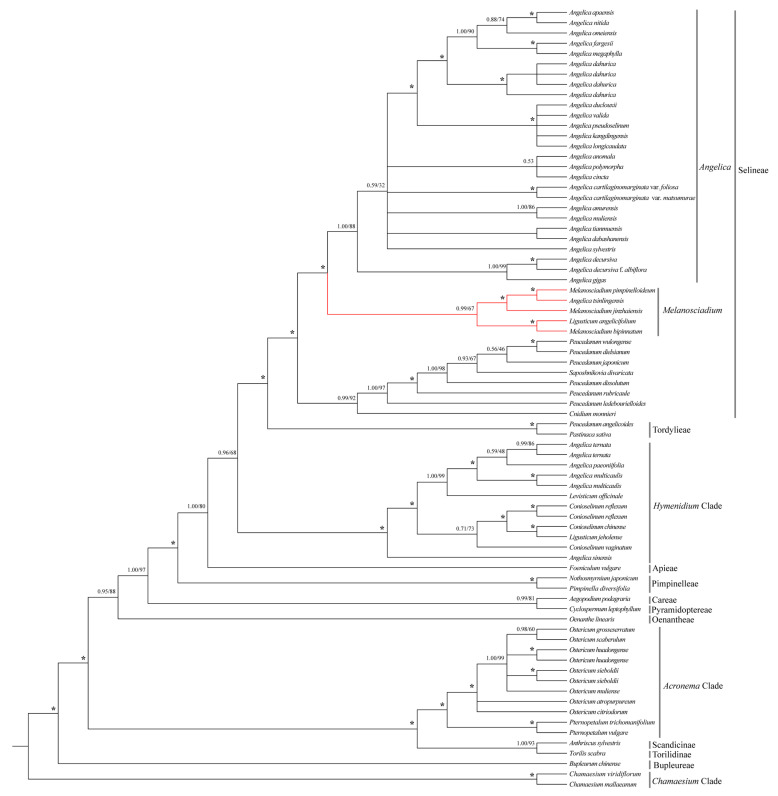
Bayesian 50% majority-rule consensus tree was inferred from partial nrDNA sequences (includes ETS, 18S rRNA, ITS 1, 5.8S rRNA, ITS 2, complete sequence, and 26S rRNA partial sequence). The tree is rooted in two species of *Chamaesium*. Numbers at nodes indicate support values obtained from Bayesian inference (BI) and Maximum likelihood (ML) methods, ***** representing the best support (100%). The red line highlights the species: *Melanosciadium jinzhaiensis*, *M. pimpinelloideum*, *M. bipinnatum*, *Angelica tsinlingensis*, and *Ligusticum angelicifolium*.

**Figure 8 plants-13-00907-f008:**
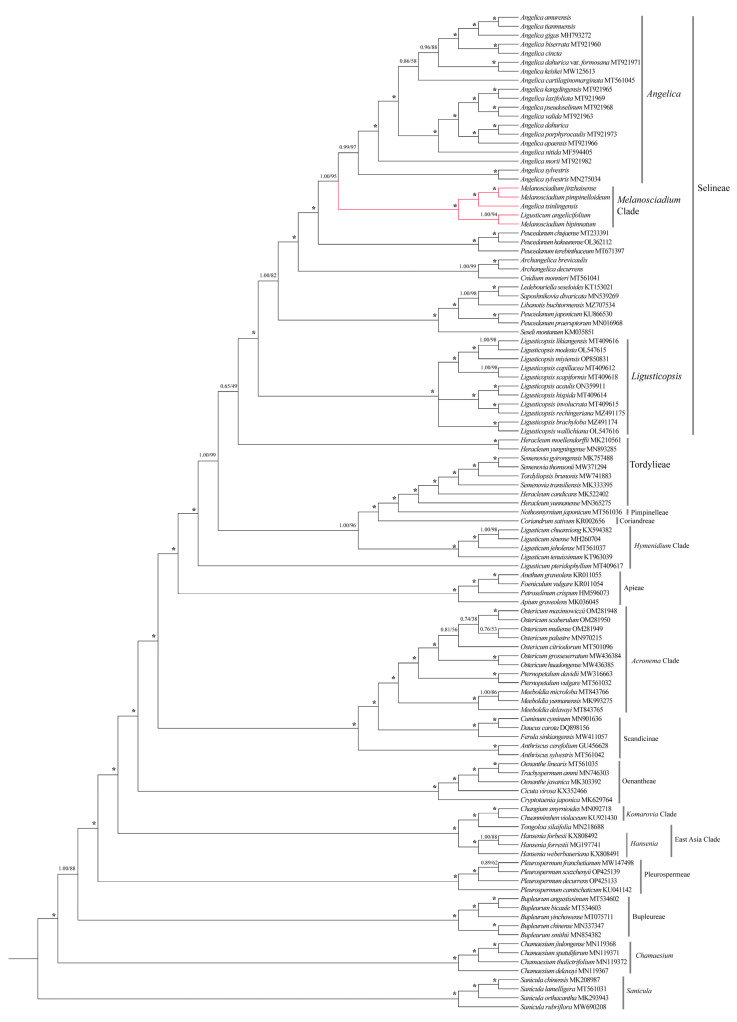
Bayesian 50% majority-rule consensus tree was inferred from cpCDS sequences. The tree is rooted in four species of *Sanicula*. Numbers at nodes indicate support values obtained from Bayesian inference (BI) and Maximum likelihood (ML) methods, ***** representing the best support (100%). The red line highlights the species: *Melanosciadium jinzhaiensis*, *M. pimpinelloideum*, *M. bipinnatum*, *Angelica tsinlingensis*, and *Ligusticum angelicifolium*.

**Figure 9 plants-13-00907-f009:**
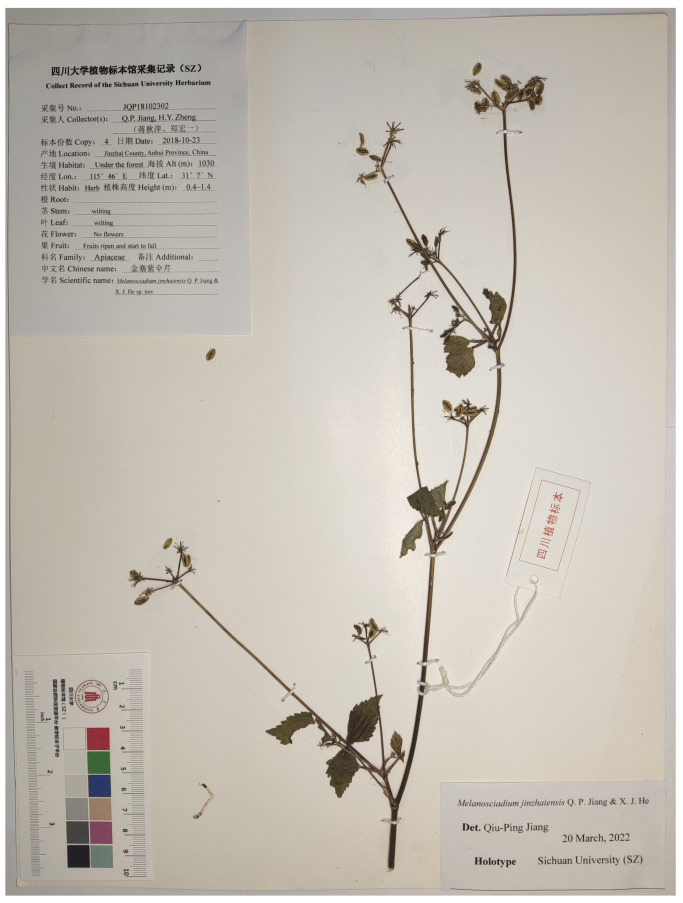
Holotype of Melanosciadium jinzhaiensis.

**Table 1 plants-13-00907-t001:** Diagnostic morphological characters of Angelica tsinlingensis, Ligusticum angelicifolium, Melanosciadium pimpinelloideum, M. bipinnatum, and M. jinzhaiensis.

Character	Taxon				
	*M. pimpinelloideum*	*M. bipinnatum*	*A. tsinlingensis*	*L. angelicifolium*	*M. jinzhaiensis*
Live form	monocarpic	monocarpic	monocarpic	monocarpic	monocarpic
Plant height (cm)	50–150	60–150	60–150	80–180	40–140
Leaf in outline (basal)	broadly ovate-triangular, 1–2-ternate	broadly ovate-triangular, 2–3-ternate	broadly ovate-triangular, 1–2-ternate	broadly ovate-triangular, 2-ternate	broadly ovate-triangular, 1–2-ternate
median leaflets (pinnae) (basal)	rhombic to narrow-ovate, 7–15 × 3–9 cm, base cuneate, margin incised-serrate, teeth mucronate, with attenuate apex	rhombic-obovate, 3–7.5 × 2–4 cm, base cuneate, margin incised-serrate, teeth mucronate, with attenuate apex	rhombic-obovate, 7–13 × 5–9 cm, base cuneate, margin incised-serrate, teeth mucronate, with attenuate apex	rhombic to oblong-ovate, 3.5–6 × 1.5–3 cm, sometimes 3-lobed, base cuneate, margin incised-serrate, teeth mucronate, with attenuate apex	short-petiolulate, rhombic-obovate, 10–15 × 5–10 cm, base cuneate; margin incised-serrate, teeth mucronate, with attenuate apex
lateral leaflets (pinnae) (basal)	ovate or longovate, 3–10 × 2–6 cm, base truncate, margin incised-serrate, teeth mucronate, apex acuminate	rhombic-obovate to longovate, 2–5.5 × 1–3 cm, base cuneate, margin incised-serrate, teeth mucronate, with attenuate apex	broad-ovate, 5–12 × 3–7 cm, base truncate, margin incised-serrate, teeth mucronate, apex acuminate	rhombic to oblong-ovate, 1–5 × 1–2 cm, sometimes 1–2-lobed, base cuneate, margin incised-serrate, teeth mucronate, with attenuate apex	broad-ovate, 7–12 × 4–7 cm, sometimes 1–2-lobed, base truncate, margin incised-serrate, teeth mucronate, apex acuminate or cuspidatus
Sheaths	not inflated to inflated	slightly inflated	not inflated	inflated	not inflated
Umbels rays	6–13, short, very unequal	14–30, very unequal	9–20, very unequal	10–30, very unequal	3–12, very unequal, short
Petals	purplish to purple; the outer not enlarged	purplish, the outer not enlarged	white, the outer slightly enlarged	white to purple, the outer not enlarged	purplish; the outer not enlarged
Calyx teeth	absent or very small	absent	absent	absent	minute
Fruit	ovoid-globose to subglobose; 3.01–4.45 × 4.16–5.64 mm	oblong-ovoid to subglobose; 2.34–4.64 × 3.28–3.93	oblong to suborbicular; 2.96–3.88 × 4.19–5.54	oblong-ovoid; 2.95–3.96 × 5.08–6.46	oblong; 2.43–3.84 × 4.03–6.68
Mericarp ribs	5 ribs; ribs prominent, ridged	5 ribs; ribs prominent, ridged	5 ribs; dorsal ribs narrow-winged, lateral ribs winged to broad-winged	5 ribs; dorsal ribs narrow-winged, lateral ribs winged	3–5 ribs; dorsal ribs narrow-winged to winged, lateral ribs winged to broad-winged
Endosperm (at commissural side)	plane to slightly concave	concave	plane to concave	slightly concave to concave	plane to slightly concave
Vittae in dorsal furrows	3–4	2–5	2–4	1–4	2–4
Vittae in commissure	4–6	6–8	4–8	4–6	6–8

**Table 2 plants-13-00907-t002:** The features of the plastid genomes of Angelica tsinlingensis, Ligusticum angelicifolium, Melanosciadium pimpinelloideum, M. jinzhaiensis, and three Angelica species (A. dahurica, A. cincta, and A. sylvestris).

Species	Size (bp)	LSC Length (bp)	SSC Length (bp)	IR Length (bp)	Number of Different Genes/Total Number of Genes	Number of Different Protein-Coding Genes (Duplicated in IR)	Number of Different tRNA Genes (Duplicated in IR)	Number of Different rRNA Genes (Duplicated in IR)	Number of Genes Duplicated in IR	GC Content (%)
*Angelica dahurica*	146,847	93,539	17,674	17,817	114/129	80 (4)	30 (6)	4 (4)	15	37.5
*Angelica cincta*	146,765	93,297	17,504	17,982	114/129	80 (4)	30 (6)	4 (4)	15	37.5
*Angelica sylvestris*	147,138	93,459	17,563	18,058	114/129	80 (4)	30 (6)	4 (4)	15	37.6
*Angelica tsinlingensis*	163,608	76,649	17,549	34,705	114/144	80 (18)	30 (7)	4 (4)	30	37.5
*Ligusticum angelicifolium*	163,798	76,900	17,464	34,717	114/144	80 (18)	30 (7)	4 (4)	30	37.4
*Melanosciadium jinzhaiensis*	164,544	76,578	17,539	35,212 (35,215)	114/144	80 (18)	30 (7)	4 (4)	30	37.4
*Melanosciadium pimpinelloideum*	164,329	76,450	17,565	35,157	114/144	80 (18)	30 (7)	4 (4)	30	37.5

## Data Availability

The data used in this study have been published and cited.

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
