# Peer review of "Phylogeny and Taxonomic Revision of the Genus Melanosciadium (Apiaceae), Based on Plastid Genomes and Morphological Evidence"

_plants, 2024, doi:10.3390/plants13060907_

Round 1

Reviewer 1 Report

Comments and Suggestions for Authors

A well-written and interesting paper.

A few points need attention.

Line 17. ‘rays’ should be ‘umbel rays’.

Lines 109-110. Clarify ‘bare-hand section’. Presumably this should be ‘hand section’, i.e. not a section cut using a microtome.

Line 152. ‘and’, not ‘abd’.

Line 221. ‘remaining’, not ‘rest’.

Line 277. ‘in the Selineae’, not ‘in Selineae’.

Line 336. ‘nature’, not ‘living’.

Comments on the Quality of English Language

A few corrections are needed, as listed in my comments.

Author Response

Response: Thanks for your suggestion, we have made modifications according to your comments.Specific can be found in: Line 18, Line 148-149, Line 192, Line 265, Line 324, Line 386.

Reviewer 2 Report

Comments and Suggestions for Authors

I strongly sympathize with the central idea of this paper. Still, unfortunately, the latter requires significant revision, first because neither Melanosciadium bipinnatum (R.H.Shan & F.T.Pu) Pimenov & Kljuykov nor M. genuflexum Pimenov & Kljuykov are included in the analyses.

Despite the significant amount of data, the taxonomic parts of the manuscript still have to be essentially improved.

The structure of the article could be much better. Perhaps to improve it, it is worth describing new species separately as the first step, publishing the plastid genome of Angelica sylvestris L. in an additional article as the next step, and only after that switching to the taxonomic treatment and phylogeny of Melanosciadium. The latter must include both M. bipinnatum and M. genuflexum. The material should be obtainable for at least the few-loci phylogeny (herbarium samples of both taxa are available).

The sampling of most of the tribes included in phylogenetic analyses must be improved.

Minor

English still requires some polishing.
The quality of images could be more accurate and must be improved.
The list of references must be updated (for example, Samigullin et al., 2022, and Cai et al., 2022 are not included).

Comments on the Quality of English Language

Moderate editing of the English language required

Author Response

Response: Thanks for your suggestion, for some reason, we prefer to write them in one manuscript. And we have revised the manuscript accordingly. We have added the recent research progress of umbelliferae in the introduction, and the list of references is updated. In this revised manuscript, we add the Ligusticopsis sequence and the sequences of Melanosciadium bipinnatum. For M. genuflexum, it's only a type specimen, there's no other record of it. Therefore, it is difficult for us to obtain its sequence. In addition, the quality of the images we also improved, we sent the high-resolution pictures to the editor.